# Risk of Major Cardiovascular Disease after Exposure to Contrast Media: A Nationwide Population-Based Cohort Study on Dialysis Patients

**DOI:** 10.3390/metabo13020266

**Published:** 2023-02-13

**Authors:** Shih-Ting Huang, Tung-Min Yu, Chia-Hsin Chen, Yun-Chung Cheng, Ya-Wen Chuang, Cheng-Hsu Cheng, Jia-Sin Liu, Chih-Cheng Hsu, Ming-Ju Wu

**Affiliations:** 1Division of Nephrology, Department of Internal Medicine, Taichung Veterans General Hospital, Taichung 40705, Taiwan; 2Department of Post-Baccalaureate Medicine, College of Medicine, National Chung Hsing University, Taichung 40227, Taiwan; 3Graduate Institute of Biomedical Sciences, School of Medicine, China Medical University, Taichung 404333, Taiwan; 4Department of Radiology, Taichung Veterans General Hospital, Taichung 40705, Taiwan; 5Department of Life Science, Tunghai University, Taichung 407224, Taiwan; 6Institute of Population Health Sciences, National Health Research Institutes, Miaoli 35053, Taiwan; 7Department of Health Services Administration, China Medical University, Taichung 40402, Taiwan; 8National Center for Geriatrica and Welfare Research, National Health Research Institiutes, Yunlin 63247, Taiwan; 9Department of Family Medicine, Min-Sheng General Hospital, Taoyuan 33044, Taiwan; 10RongHsing Research Center for Translational Medicine, National Chung Hsing University, Taichung 40227, Taiwan; 11School of Medicine, Chung Shan Medical University, Taichung 40201, Taiwan

**Keywords:** iodinated contrast media, cardiovascular disease, dialysis, inflammasome

## Abstract

Contrast associated kidney injury is caused by side effects of iodinated contrast media (ICM), including inflammation. Chronic inflammation among dialysis patient contributes to atherosclerosis, which leads to simultaneous conditions of the kidney, brain, and vasculature. Data to investigate the pathologic effects of ICM on cardiovascular complications in dialysis patients are lacking. Dialysis patients who had been exposed to ICM from computed tomography (ICM-CT) were allocated as the ICM-CT cohort (N = 3751), whereas dialysis patients without ICM exposure were randomly allocated as the non-ICM cohort (N = 17,196). Furthermore, 540 pairs were selected for analyses through propensity score-matching in terms of age, sex, comorbidities, dialysis vintage, and index date. During a median follow-up of 10.3 years, ICM-CT cohort had significantly higher risks in the following, compared with non-ICM cohort: all-cause mortality (adjusted hazard ratio [aHR], 1.36; 95% confidence interval [CI], 1.26–1.47), cardiovascular events (aHR,1.67; 95% CI, 1.39–2.01), acute coronary syndrome (adjusted HR: 2.92; 95% CI, 1.72–4.94), sudden cardiac arrest (aHR, 1.69; 95% CI, 0.90–3.18), heart failure (aHR, 1.71; 95% CI,1.28–2.27), and stroke (aHR, 1.84; 95% CI,1.45–2.35). The proinflammatory ICM is significantly associated with an increased risk of major cardiovascular events in patients on dialysis.

## 1. Introduction

Cardiovascular disease (CVD) is a major cause of morbidity and mortality in patients on dialysis [1]. It is present in more than 50% of patients on dialysis, and the relative risk of cardiovascular death in these patients is 10 to 20 times higher compared with the general population [2]. Diabetes, hypertension, and end-stage renal disease (ESRD) are independent CVD risk factors [3,4]. Their higher risk of CVD may be related to the perplexity of the uremic milieu. In addition to traditional CVD factors, uremic factors such as chronic inflammation, hyperhomocysteinemia, altered lipid metabolism, and hyperparathyroidism contribute to their CVD risks [5,6].

Chronic inflammation is considered a key nontraditional CVD risk in populations on dialysis. Chronic inflammation is prevalent in patients on dialysis and causes premature and progressive atherosclerotic CVD [7]. Because of the lack of antioxidant enzymes associated with renal insufficiency and the exacerbating effect of dialysis on directly increasing plasma reactive oxygen species (ROS) levels, the resulting oxidative stress contributes to endothelial damage and inflammation [8]. Asymmetric dimethylarginine (ADMA) likely accelerates the progression of endothelial dysfunction because of the accumulation of endogenous NOS inhibitors. A high level of ADMA in patients with ESRD is a predictor of their CVD and death [9]. Both pharmacological and nonpharmacological interventions are devoted to reducing chronic systemic inflammation in patients on dialysis. However, the evidence supporting outcome improvement is still limited [10].

Iodinated contrast media (ICM) are contrast agents that contain iodine atoms. ICM is used for X-ray-based imaging such as computed tomography (CT) and angiography. Exposure to ICM is associated with detrimental pathologic effects, including anaphylactoid reactions, nephrotoxicity, and neurotoxicity. In addition, recent study revealed that ICM is associated with an increased dementia risk in the general population [11]. All classes of iodinated contrast media have complex vasomotor effects on several arterial districts, and that atherosclerosis has an impact on the type of the contrast-induced coronary vasomotor reaction [12]. Until recently, most studies have focused on immediate effects of intravenous ICM during cardiac angiography. In a double-blind randomized study, compare the hemodynamic changes induced by non-ionic low osmolar (Lomeprol 350, IO) and ionic high osmolar (Diatrizoat 370, DIA), DIA caused a drop in diastolic aortic pressure after 10 to 15 sec and a bradycardia in the first five sec, in contrast to IO after selective coronary angiography. These effects can be explained by a cardiodepressive action of contrast agents on left ventricular function, an increase in circulating volume and a reduced peripheral vessel resistance [13]. In a laser argon-induced thrombosis model in mesenteric microvessels, Iopamidol and iohexol demonstrated prothrombotic properties by inducing a significant rise in the number of emboli and the duration of embolization [14]. Exposure to ICM can exacerbate hyperthyroidism in elderly patients with underlying thyroid disease, and that can lead to serious heart failure complication secondary to ischemic events [15]. However, previous CIM-related cardiovascular studies had limited case numbers, and the long-term effects of intravenous ICM-CT exposure on the cardiovascular system in the dialysis population have not been well studied.

Intravenous iodinated contrast media are potent inflammation inducers. In practice, the use of ICM-based CT (ICM-CT) represents a majority of ICM exposures. In patients on dialysis, prolonged retention of ICM in the extracellular space likely induces sustained systemic inflammation. Here, we aimed to evaluate in these patients whether exposure to intravenous ICM-CT increases risks of all-cause mortality and major adverse cardiovascular events (MACEs).

## 2. Materials and Methods

### 2.1. Data Source

In this retrospective cohort study, we analyzed database obtained from the Registry for Catastrophic Illness Patients (CIPR), which consists of disease categories including ESRD, with verifying chronic dialysis and the irreversible status by two nephrologists. The data represented a subset of the National Health Insurance Research Database (NHIRD) covering a period from 1 January 2000, to 31 December 2016 (National Science Council—Taiwan, RRID: SCR_011434). The NHIRD, with comprehensive records of claims relating to outpatient and inpatient medical care, enrolled data from 23 million Taiwanese people as in 2015 [16,17]. The RCID comprises information on the registry of beneficiaries, diagnostic codes, medical procedures, and prescription data. All data extracted from the NHIRD were deidentified to protect privacy.

Definitions of diseases are based on the International Classification of Diseases, 9th Revision, Clinical Modification for cases diagnosed before 1 January 2016 and the International Classification of Diseases, 10th Revision, Clinical Modification for cases diagnosed afterwards. The list of ICD-9 and ICD-10 codes were used to define the inclusion criteria for patients on dialysis, outcome events, and comorbidities as shown in Appendix A.

### 2.2. Ethics Statement

Our study met the conditions for waiving patient informed consent and was approved by the Institutional Review Board of the National Health Research Institutes (EC1060704-E).

### 2.3. Study Design

Figure 1 shows the study design and flowchart for patient selection. We identified 243,156 newly diagnosed patients with newly diagnosed ESRD (ICD-9 CM: 586; ICD-10: N18.6; and were registered in the RICD database) and the patients all underwent chronic hemodialysis or peritoneal dialysis for more than 3 months. Patients on dialysis were excluded under these conditions: (a) ICM-CT applied before the initiation of dialysis, (b) having missed information, or (c) receiving renal transplant during the observation period. For each patient with ICM exposure, the index date was the date of first exposure to ICM-CT. For each patient without ICM exposure, the index date was randomly assigned, based on their age, gender, and dialysis vintage relative to patients with ICM-CT exposure. Patients were also excluded if they: (a) were aged >100 or <20 years; (b) underwent CT or ICM-CT due to brain or cardiovascular diseases before the index date to avoid bias; (c) had prior hospitalization for acute myocardial infarction (AMI), acute coronary syndrome (ACS), sudden cardiac arrest (SCA), ischemic stroke, heart failure, or cancer; or (d) died within 90 days of the start of dialysis treatment. Finally, we analyzed 3751 patients in the ICM-CT cohort and 17,196 patients in the non-ICM cohort with matching age, gender, and dialysis vintage. To match for balance and pair for heterogeneity among study cohorts, we further performed propensity score matching at a 1:1 ratio (*n* = 540).

### 2.4. Independent Variables

Demographic information and preexisting comorbidities that affect outcomes, such as age, gender, premium level, urbanization, hypertension, diabetes, hyperlipidemia, dysrhythmia, chronic obstructive pulmonary disease (COPD), and liver disease were investigated. The Charlson comorbidity index score was used as the composite score of comorbidities at baseline. The premium levels and residence urbanization as the sociodemographic factors and the according to the last insurance recode of patients before initial dialysis. The preexisting comorbidities were defined as diagnoses appeared at least 3 times on outpatients’ records and at least 1 time on inpatients’ records. Dialysis vintage of a patient was calculated from the date of regular dialysis to the index date. Data were listed regarding concurrent medications (i.e., hypoglycemic agent, antihypertensive agent, diuretic, statin, and nonsteroidal anti-inflammatory drug [NSAID]; which was used at least 3 months upon the initiation of dialysis) with details shown in Appendix A.

### 2.5. Outcome Measurement

The primary endpoint was all-cause mortality, and deaths were verified in the national death registry. The secondary endpoint was hospitalization for newly diagnosed MACEs (ACS, AMI, SCA, heart failure, stroke) during the year after the index date. MACEs outcomes were followed independently as disjointed events.

The accuracy of the disease diagnoses using ICD coding system in the NHIRD database has been validated [18,19,20]. The follow-up was from the index date to the occurrence of an outcome diagnosis. Patients were censored at the date of death, withdrawal from the health insurance system, or on 31 December 2016, whichever came first.

### 2.6. Statistical Analyses

Baseline characteristics were compared between study cohorts using either a two-tailed *t*-test or χ2 test, whichever was appropriate. Categorical variables were presented as number (percent), and continuous variables were presented as median or mean ± standard deviation. The propensity score was estimated using logistic regression to determine the probability of exposure to intravenous ICM-CT, according to the baseline variables (i.e., age, sex, dialysis vintage, urbanization level, premium level, comorbidity, Charlson comorbidity index, and prescription medication). The cumulative incidence of outcome was computed using the Kaplan–Meier method, and log rank test was used to verify the equality of survivor functions between study cohorts. Multivariable Cox proportional hazard regression models were used to determine effects of ICM exposure on the outcome risks in the cohorts. Results were expressed as hazard ratios (HRs). Adjusted HRs (aHRs) and 95% confidence intervals (CIs) for MACEs and all-cause mortality were further analyzed according to patients’ characteristics. The log minus log plots of all categorical variables did not present the cross. This did not violate the assumption of hazard proportional in survival regression analysis. Stratified analyses based on age, gender, dialysis vintage, and comorbid conditions were used to evaluate and control for confounding factors. We used the proportional subdistribution hazard models developed by Fine and Gray [21] and the cause-specific adjusted model [22] to modify competing risks. The fine and gray competing risk model was a subdistribution function and was using a cumulative risk function for an estimate of the event risk that interrupts because of the competing risk event incidence. Statistical significance was set at two-sided *p* values < 0.05. All statistical analyses were performed using STATA (Version 15.1, College Station, TX, USA) and SAS (Version 9.4; SAS Institute, Inc., Cary, NC, USA).

## 3. Results

### 3.1. Baseline Characteristics

Table 1 displays the distribution of age, sex, and comorbidities of the ICM-CT and non-ICM CT cohorts. The mean age of dialysis in the ICM-CT group was 57.2 (±15.7) years and the non ICM CT group was 62.7 (±12.8) years before propensity score matching (PSM). After PSM, the mean age of dialysis in the ICM-CT group was 63.1 (±16.7) years and the non ICM CT group was 55.0 (±15.6) years. Before PS-matching, the ICM-CT cohort included patients who were predominantly younger (52.7 years), female (54.5%), and had hypertension (78.9%), diabetes (40.4%), hyperlipidemia (24%), dysrhythmia (9.0%), COPD (21.8%), and liver disease (26.9%). The mean exposure episodes in the ICM-CT exposure cohort were 2.73 (SD, 2.29) during the follow-up period. In contrast, patients in the non-ICM cohort were predominantly older (mean: 62.7 years), with lower premiums (57.9% paid premiums of <NT$22,000), fewer comorbidities (CCI score: 3.5), and they had a longer mean dialysis vintage (3.0 years). The mean follow-up period for patients developing outcome in the adjusted model was 5.6 ± 4.6 years for the ICM-CT, and 11.3 ± 5.5 years for the non-ICM cohort. In the PS-matched model, the ICM-CT and non-ICM cohorts had similar baseline characteristics.

### 3.2. Primary Outcome

The all-cause mortality rate was much higher in the ICM-CT cohort compared with the non-ICM cohort (96.49 vs. 12.3 per 1000 person-years) (Table 2). In the multivariate model, the crude HR (cHR) of mortality was 5.29 fold higher in the ICM-CT cohort (95% CI, 4.98–5.61) than that in controls. Notably, 347 patients in the ICM-CT cohort had developed composite MACEs at an incidence rate of 29.65 per 1000 person years. In the adjusted model, the ICM-CT cohort had significantly higher risks of the following: all-cause mortality (aHR, 1.36; 95% CI, 1.26–1.47), hospitalization for composite MACEs (aHR, 1.67; 95% CI, 1.39–2.01), ACS (aHR, 2.92; 95% CI, 1.72–4.94), SCA (aHR, 1.69; 95% CI, 0.90–3.18), heart failure (aHR, 1.71; 95% CI, 1.28–2.27), and stroke (aHR, 1.84; 95% CI, 1.45–2.35). In the propensity score–matched adjusted model, the risk of ICM-CT exposure remained higher with respect to all-cause mortality (aHR,1.66; 95% CI, 1.35–2.04), MACEs (aHR,3.90; 95% CI, 2.08–7.33), SCA (aHR,6.33; 95% CI, 1.01–39.5) and heart failure (aHR,3.13; 95% CI, 1.29–7.60). The Kaplan–Meier curves indicated that patients in the ICM-CT cohort had significantly higher risks of both all-cause mortality and MACEs relative to the non-ICM cohort during the study period (Figure 2).

### 3.3. Sensitivity and Subgroup Analyses

Subgroup analyses (Figure 3) revealed that the ICM-CT cohort, compared with the non-ICM cohort, had a significantly higher risk of mortality regardless of range of ages on initiation of dialysis, gender, dialysis vintage (>1 year), and diabetic status. Patients in the ICM-CT cohort with no concomitant hypertension and hyperlipidemia also had a higher risk of mortality compared with the non–ICM cohort. Similarly, these patients in the ICM-CT cohort also had a significantly higher risk of MACEs regardless of age, gender, dialysis vintage, diabetic status, and ACEi/ARB. Additionally, those patients in the ICM-CT cohort without hypertension and hyperlipidemia also had a significantly higher risk of MACEs. Notably, the ICM-CT cohort who did not use hypoglycemic agents, angiotensin-converting-enzyme inhibitors (ACEi) and angiotensin II Receptor blockers (ARBs), beta-blockers, CCBs, or statins still showed significantly higher risks in mortality and MACEs compared with similar nonusers in the non-ICM cohort.

In the subdistribution of the competing risk–adjusted model, MACE risks were consistently higher in the ICM-CT cohort (aHR, 1.79; 95% CI, 1.46–2.20) than in the non-ICM cohort (Table 3). The risk of ACS and AMI was 3.08 and 4.02 times higher, respectively, in the ICM-CT-exposure group than in the non-ICM group. Risks of SCA were similar in the two study groups.

## 4. Discussion

Our study may be the first population-based study to evaluate the long-term risks of cardiovascular complications and mortality after intravenous ICM exposure on chronic dialysis patients. The study verifies that the ICM-CT cohort had an increased risk of all-cause mortality and MACEs compared with non-ICM exposures. Our results indicated that dialysis patients who received intravenous ICM exposures—a group of patients traditionally considered to be safe under ICM exposures—have clinically relevant long term major cardiovascular risks.

Intravenous iodinated contrast media are widely used with computed tomography to evaluate disease and treatment response. ICM have been denied or delayed in patients with chronic kidney disease due to perceived risks of contrast-induced acute kidney injury. In bench studies, ICM induce not only hyperviscosity but also inflammation in renal tubules and peritubular capillaries. In patients on dialysis, ICM-enhanced CT is generally considered safe; however, long term safety from repeated ICM exposure remains unclear. We hypothesized that ICM, as a middle molecule weight particle, might not been fully dialyzable, and tissue redistribution can cause inflammation and subsequent complications.

In the dialysis population, contrast media are believed to be removed from blood through hemodialysis or peritoneal dialysis. Pharmacokinetic studies of water-soluble ICM reveal an initial distribution phase and a subsequent elimination phase with respect to renal clearance [23]. However, the extent of ICM diffusion into intracellular space (e.g., cardiomyocytes, neurons, and microvasculature) had not been assessed. Although the necessity of immediate dialysis after intravascular injection of ICM was studied in patients on chronic hemodialysis, no evidence has yet been provided to support its effectiveness in preventing CA-AKI [24,25].

Vascular diseases, including those affecting the coronary, cerebral, renal, and retinal vasculature, are associated with a predisposition to vascular injury due to their similarities in development, morphology, and pathophysiology [26,27]. Inflammation is a key driver of atherosclerosis. The recruitment of monocytes and their differentiation into macrophages after noxious stimuli, including ICM, were demonstrated in the formation of atherosclerotic plaque [28,29].

Novel pathways of inflammation are linked to the development of atherosclerosis [30,31,32]. Somatic mutations in the gene encoding the TET2 enzyme promote clonal hematopoietic expansion and accelerate atherosclerosis in mice with hyperlipidemia [30]. TET0-deficient animals show an increased secretion of cytokines and chemokines by macrophages via the expression of the Nod-like receptor pyrin containing 3 (Nlrp3) inflammasome and interleukin-1β. These results are related to mechanisms of exacerbated atherosclerosis [31,32]. Nlrp-3 is an intracellular sensor activated by numerous stimuli, including self- and foreign- derived activators and pathogens. As in 2018, pivotal studies have highlighted the role of inflammation in the pathogenesis of CA-AKI. Lau and Shen et al., reported that contrast media activates the canonical Nlrp3 inflammasome in macrophages, contributing to CA-AKI through regulating inflammation [33,34]. Furthermore, Nlrp3-deficient mice display fewer epithelial cell injury and inflammation in their kidneys. Results indicate that contrast-induced inflammation mediated by Nlrp-3 inflammasome is an integral component of kidney injury. Nlrp3 also has canonical and noncanonical roles in models for other kidney disease [35,36,37].

Regarding immediate effects of intravenous ICM on the cardiovascular system, Pagny et al., reported a cardio-depressive effect on the left ventricular function together with electrophysiological changes resulted from the inherent pharmacologic properties of ICM (i.e., osmolality, viscosity, and chemotoxicity) [38]. In our cause-specific adjusted hazard model, the mean number of episodes of exposure to intravenous ICM in the ICM-CT cohort was 2.73 (SD, 2.29) during the follow-up period, and the risk of all MACEs in the ICM-CT cohort, particularly ACS (aHR, 3.08; 95% CI, 1.87–5.06) and AMI (aHR, 4.02; 95% CI,2.11–7.69), was significantly higher compared with the non-ICM cohort (Table 3). Accordingly, we inferred that multiple exposures with contrast media have enduring proinflammatory effects on microvascular and macrovascular tissues.

Stroke is a major complication of patients on dialysis [39]. Our results indicated that the ICM-CT cohort had a 96% increase in the risk of stroke relative to the non-ICM cohort (Table 3). Although, as endpoints, we did not include contrast-induced encephalopathy (CIE), which can be considered as a manifestation of contrast-media-induced microvascular disease. CIE is an acute and reversible neurological disturbance directly attributed to intra-arterial administrations of ICM following angiography [40]. Imaging findings of CIE may mimic subarachnoid hemorrhage or cerebral ischemia. The putative mechanism involves the disruption of the blood–brain barrier, direct neuronal injury, and transient vasoconstriction following intravenous ICM [41]. Patients on dialysis who have undergone cerebral arteriography are susceptible to CIE due to the delayed excretion of contrast media [42]. These findings provided us indirect evidence of the cerebral microvascular damage caused by ICM.

We hypothesized that the activation of Nlrp3 inflammasome plays a role in atherosclerosis, and the contrast media serves as its activator. Future research works are needed to test this hypothesis. To reduce intravenous ICM exposure, novel CT-based dose-reduction techniques, such as photon-counting detectors, offer promising results with better contrast-to-noise ratio and spatial resolution [43].

### Limitations

Our results should be interpreted with caution due to the inherent limitations of any retrospective observational cohort study. First, this study presents the main methodological limitation as the main clinical diagnosis was obtained retrospectively from a Health Insurance Research Database. Like with all claims databases, there have been some validity concerns of studies using the NHIRD, such as the accuracy of diagnosis codes and issues around unmeasured confounders. Second, before 2016, the International Classification of Diseases, Ninth Edition, Clinical Modification (ICD-9-CM) was used for recording diagnosis in Taiwan’s National Health Insurance Research Database, and the Tenth Edition (ICD-10) has been used since 2016. This could be an inherent limitation. However, several validation studies have been performed to evaluate the validity of diagnosis codes in the NHIRD. Most of the validated diagnosis codes are common conditions or severe diseases with modest to high sensitivity and positive predictive values [19]. Third, because our study is not a prospective randomized controlled trial, we could not establish a cause-and-effect relationship accordingly. Fourth, our findings were derived from statistical inference based on regression models with adjustments. Despite minimal baseline imbalance between the study cohorts, we meticulously dealt with sensitivity and validation analyses. At last, the claim database was limited because administrative data, smoking status, body mass index, laboratory data, exact dose, and detailed ICM osmolarity may be helpful, but is not available in our database. However, the diagnosis of end stage renal disease is validated since medical reimbursement of dialysis care is applied after quarterly reporting longitudinal laboratory data (such as measurement of dialysis clearance, hemoglobin, albumin, intact-PTH data) by attending physicians. Thus, we applied comorbidities and dialysis vintage as a proxy for underlying disease burden, which is an important predisposing factor for mortality and cardiovascular outcome.

## 5. Conclusions

ICM-CT exposure was significantly and independently associated with major cardiovascular diseases and all-cause mortality in the dialysis population. To reduce intravenous ICM exposure, novel CT-based dose-reduction techniques are encouraged in patients on dialysis. Bench studies are required to validate the in vitro effect of ICM on myocardial and coronary vascular cells in experimental models.

## Figures and Tables

**Figure 1 metabolites-13-00266-f001:**
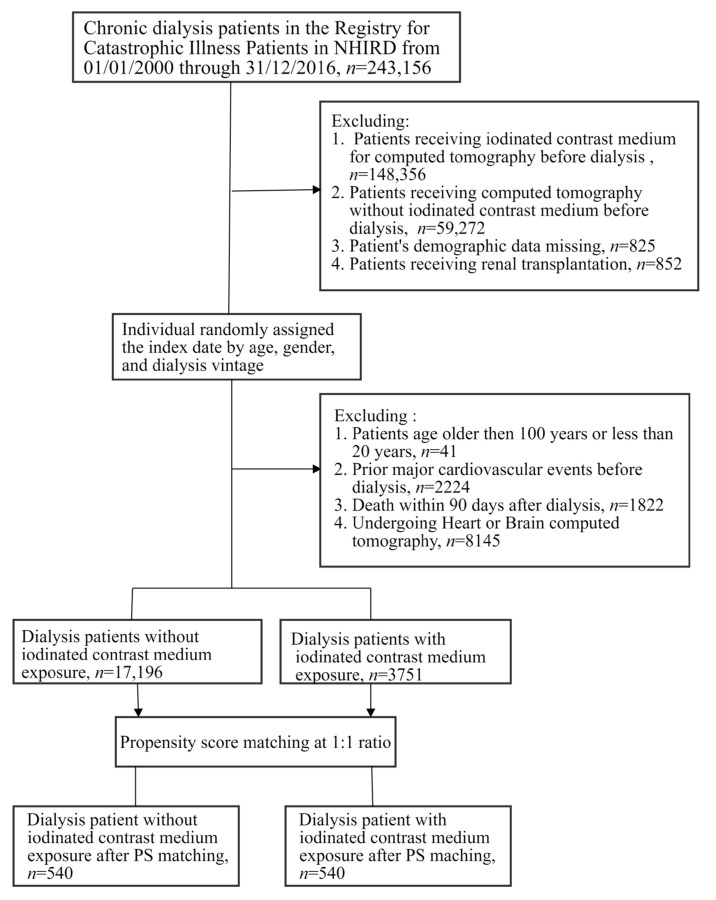
Study design and patient selection flowchart.

**Figure 2 metabolites-13-00266-f002:**
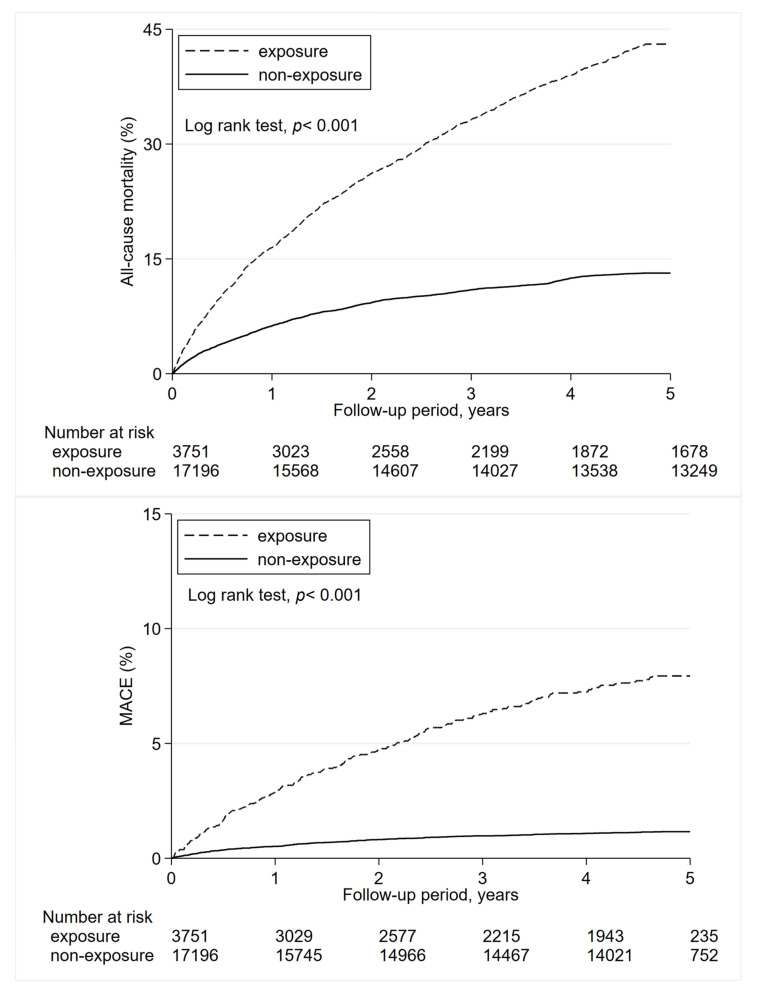
Cumulative risk curves of all-cause mortality and major adverse cardiovascular events for the study cohorts with ICM-CT and non-ICM-CT users.

**Figure 3 metabolites-13-00266-f003:**
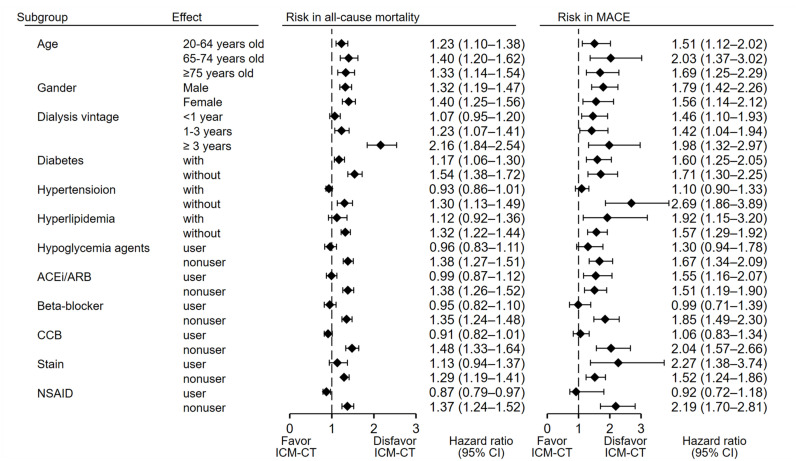
Sensitivity and subgroup analyses of comparative risks for all-cause mortality and major adverse cardiovascular events of the ICM-CT exposure cohort compared to the non ICM-CT cohort.

**Table 1 metabolites-13-00266-t001:** Baseline characteristics of study cohorts before and after propensity score matching process.

	Before PSM ^†^			After PSM		
Variable	ICM-CT Group N = 3751	Non-ICM Group N = 17,196	*p* Value	ICM-CT Group N = 540	Non-ICM Group N = 540	*p* Value
Age group						
Mean(SD)	57.2 (15.7)	62.7 (12.8)	<0.001	63.1 (16.7)	55.0 (15.6)	<0.001
Gender			0.045			<0.001
Male	1708 (45.5)	8140 (47.3)		241 (44.6)	309 (57.2)	
Female	2043 (54.5)	9056 (52.7)		299 (55.4)	231 (42.8)	
Premium level (NT$)			0.33			0.20
<22,000 or low income	2126 (56.7)	9956 (57.9)		285 (52.8)	283 (52.4)	
45,000–22,000	1397 (37.2)	6265 (36.4)		228 (42.2)	216 (40.0)	
>45,000	228 (6.1)	975 (5.7)		27 (5)	41 (7.6)	
Urbanization			<0.001			0.74
Highly	1006 (26.8)	4844 (28.2)		141 (26.1)	154 (28.5)	
Median	1808 (48.2)	7635 (44.4)		261 (48.3)	259 (48)	
Township	510 (13.6)	2570 (14.9)		72 (13.3)	70 (13)	
Rural area	427 (11.4)	2147 (12.5)		66 (12.2)	57 (10.6)	
Comorbidity						
Hypertension	2961 (78.9)	3743 (21.8)	<0.001	363 (67.2)	416 (77)	<0.001
Diabetic	1516 (40.4)	5288 (30.8)	<0.001	236 (43.7)	190 (35.2)	0.004
Hyperlipidemia	902 (24)	1248 (7.3)	<0.001	131 (24.3)	127 (23.5)	0.78
Dysrhythmia	336 (9.0)	281 (1.6)	<0.001	27 (5.0)	24 (4.4)	0.67
COPD	816 (21.8)	722 (4.2)	<0.001	68 (12.6)	69 (12.8)	0.93
Liver disease	1008 (26.9)	938 (5.5)	<0.001	89 (16.5)	79 (14.6)	0.40
CCI scores						
Mean(SD)	4.3 (1.8)	3.5 (1.5)	<0.001	3.4 (1.2)	3.5 (1.4)	<0.001
Dialysis vintage, years	2.8 (3.1)	3.0 (3.4)	0.013	2.2 (2.8)	2.6 (3)	<0.001
Concurrent medications						
Hypoglycemia agents	1432 (38.2)	1556 (9.0)	<0.001	95 (17.6)	73 (13.5)	0.063
ACEI/ARB	1240 (33.1)	1249 (7.3)	<0.001	147 (27.2)	139 (25.7)	0.58
Beta blocker	1899 (50.6)	1887 (11)	<0.001	131 (24.3)	105 (19.4)	0.06
CCB	1092 (29.1)	1022 (5.9)	<0.001	203 (37.6)	159 (29.4)	0.005
Diuretics	825 (22)	878 (5.1)	<0.001	114 (21.1)	76 (14.1)	0.002
Statins	1432 (38.2)	1556 (9)	<0.001	77 (14.3)	95 (17.6)	0.13
NSAIDs	1577 (42)	1453 (8.4)	<0.001	143 (26.5)	131 (24.3)	0.40
Follow-up time, years	5.6 (4.6)	11.3 (5.5)	<0.001	3.3 (3.7)	4.9 (4.7)	<0.001

^†^ PSM: propensity score matching; SD, standard deviation. ICM-CT: iodinated contrast media of computed tomography; non-ICM: non-exposure to iodinated contrast media; NT$: USD1 is equivalent to approximately NT$30. Abbreviations: ACEI, Angiotensin Converting Enzyme inhibitors; ARB, Angiotensin receptor blockers; CCB, calcium channel blockers; CCI, Charlson comorbidity index; COPD: Chronic Obstructive Pulmonary Disease; NSAID, non-steroidal anti-inflammatory drugs.

**Table 2 metabolites-13-00266-t002:** Incidence and hazard ratios of all-cause mortality and MACEs in patients with ICM-CT exposures compared to those without, by type of matching.

	ICM-CT Cohort N = 3751		Non-ICM Cohort N = 17,196		Crude Model		Adjusted Model ^†^		After Propensity Matched Adjusted Model	
Outcome	Event	IR	Event	IR	cHR (95% CI)	*p* Value	aHR (95% Confidence Interval)	*p* Value	Propensity Score–Matched HRs (95% Confidence Interval)	*p* Value
All-cause mortality	2042	96.49	2396	12.3	5.29 (4.98–5.61)	<0.001 ***	1.36 (1.26–1.47)	<0.001 ***	1.66 (1.35–2.04)	<0.001 ***
Hospitalization for										
MACE	347	29.65	390	5.46	5.54 (4.76–6.44)	<0.001 ***	1.67 (1.39–2.01)	<0.001 ***	3.90 (2.08–7.33)	<0.001 ***
ACS	64	5.29	45	0.63	9.98 (6.60–15.1)	<0.001 ***	2.92 (1.72–4.94)	<0.001 ***	4.72 (0.48–46.2)	0.18
AMI	46	3.8	26	0.36	12.1 (7.20–20.3)	<0.001 ***	3.94 (2.01–7.70)	0.10	-	
SCA	31	2.56	30	0.42	5.43 (3.28–8.99)	<0.001 ***	1.69 (0.90–3.18)	<0.001 ***	6.33 (1.01–39.5)	0.048 *
Heart failure	153	12.81	189	2.64	6.08 (4.82–7.67)	<0.001 ***	1.71 (1.28–2.27)	<0.001 ***	3.13 (1.29–7.60)	0.011 *
Stroke	165	13.86	161	2.24	5.20 (4.18–6.47)	<0.001 ***	1.84 (1.45–2.35)	<0.001 ***	-	

N, case number; IR, incidence rate, per 1.000 person years; IR: Incidence rate, per 1.000 person years, 95% CI: 95% confidence interval; cHR, crude hazard ratio; aHR: adjusted hazard ratio; *, *p* < 0.05; *** *p* < 0.001; MACE, major adverse cardiovascular events; ACS, acute coronary syndrome; AMI, acute myocardial infarction; SCA, sudden cardiac arrest. ^†^ Model adjusted for age, sex, dialysis vintage, urbanization level, premium level, comorbidities, Charlson comorbidity index, and all prescription medications.

**Table 3 metabolites-13-00266-t003:** Adjusted subhazard ratio (aSHR) of MACEs in the ICM-CT cohort using competing risk regression models.

Outcome	The Subdistribution Competing Risk Adjusted Hazard Model ^†^		
	aSHR (95% Confidence Interval)		*p* Value
	Non-ICM Cohort N = 17,196	ICM-CT Cohort N = 3751	
Hospitalization for			
MACE	1 (Reference)	1.79 (1.46–2.20)	<0.001 ***
ACS	1 (Reference)	3.14 (1.77–5.56)	<0.001 ***
AMI	1 (Reference)	3.98 (1.87–8.48)	<0.001 ***
Sudden cardiac arrest	1 (Reference)	1.39 (0.71–2.71)	0.33
Heart failure	1 (Reference)	1.75 (1.29–2.38)	<0.001 ***
Stroke	1 (Reference)	2.01 (1.52–2.66)	<0.001 ***

^†^ The model was adjusted for age, sex, dialysis vintage, urbanization level, premium level, comorbidities, Charlson comorbidity index, and all prescription medications. *** *p* < 0.001; MACE, major adverse cardiovascular events; ACS, acute coronary syndrome; AMI, acute myocardial infarction; SCA, sudden cardiac arrest. N, case number; IR, incidence rate, per 1.000 person years.

## Data Availability

The datasets presented in this article are not readily available because the data underlying this article are stored in the National Health Insurance Research Database (NHIRD) of Taiwan. It can only be accessed via direct approval via the NHIRD. Requests to access the datasets should be directed to sgazn.tw@gmail.com.

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
