# Peer review of "Risk of Major Cardiovascular Disease after Exposure to Contrast Media: A Nationwide Population-Based Cohort Study on Dialysis Patients"

_metabolites, 2023, doi:10.3390/metabo13020266_

Round 1
Reviewer 1 Report
The authors have investigated the risk of cardiovascular diseases after exposure to contrast media. Overall, the methodologies used in this study are appropriate for the aim of the study. The results are interesting and well discussed. The article can be accepted after some minor corrections:
I have the following comments:
1. The introduction section must be revised by including relevant recent literature and their limitation with respect to the present work. For example:
Diepenbroek SM, de Jonghe A, van Rees C, Seebus E. Heart failure as a serious complication of iodinated contrast-induced hyperthyroidism: case-report. BMC Endocr Disord. 2021 Oct 20;21(1):207. doi: 10.1186/s12902-021-00870-y.
2. The texts in the figure are hard to read. For better readability, the authors must increase the text size in the figure and legends.
Author Response
- The introduction section must be revised by including relevant recent literature and their limitation with respect to the present work. For example:
Diepenbroek SM, de Jonghe A, van Rees C, Seebus E. Heart failure as a serious complication of iodinated contrast-induced hyperthyroidism: case-report. BMC Endocr Disord. 2021 Oct 20;21(1):207. doi: 10.1186/s12902-021-00870-y.
Answer: Thanks for your comment that fulfill our knowledge about how ICM also lead to hyperthyroidism and subsequent cardiovascular complications. We’ve added “Exposure to ICM can exacerbate hyperthyroidism in elderly patients with underlying thyroid disease, and that can lead to serious heart failure complication secondary to ischemic events” in the third paragraph in the introduction section. The new reference has been added into [15].
- The texts in the figure are hard to read. For better readability, the authors must increase the text size in the figure and legends.
Answer: Thanks for your comment, we’ve increased the figure size to 600DPI for each figure. For figure 3 and its legend, we added descriptions for clarification.

Reviewer 2 Report
Dear authors.
Your paper is of great interest to clinicians and patients related to dialysis and contrast media. The results presented by you are of great help for decision-making.
The introduction is very well addressed, having a progressive flow of ideas that leads to the hypothesis, or objective of the study. The methodology is well described, with emphasis on bioethics protocol and statistical analysis. The only thing I can comment on in this section is that the criteria for chronicity of dialysis are not described, what the authors consider to be chronic dialysis and the bibliographic support for this point. What was the criterion used for its definition?. It is mentioned in figure 3 but is not easily accessible to a reader. Perhaps in this section or the discussion, to see what the age of dialysis initiation is, and if it has any impact on the effects of dialysis on the patient, some references address this, but it would be a good point for discussion.
The results are set out and described in a great way. And it is in this part that I have a couple of comments.
Great discussion, and a notable research limitations section.
Author Response
The introduction is very well addressed, having a progressive flow of ideas that leads to the hypothesis, or objective of the study. The methodology is well described, with emphasis on bioethics protocol and statistical analysis. The only thing I can comment on in this section is that the criteria for chronicity of dialysis are not described, what the authors consider to be chronic dialysis and the bibliographic support for this point. What was the criterion used for its definition?. It is mentioned in figure 3 but is not easily accessible to a reader. Perhaps in this section or the discussion, to see what the age of dialysis initiation is, and if it has any impact on the effects of dialysis on the patient, some references address this, but it would be a good point for discussion.
Answer: Thanks for your comment.
- We’ve added the criteria of chronic dialysis in the “material and method” section. In the data source paragraph, we added that “In this retrospective cohort study, we analyzed database obtained from the Registry for Catastrophic Illness Patients (CIPR), which consists of disease categories including ESRD, with verifying chronic dialysis and the irreversible status by two nephrologists.”. In the study design paragraph, we added “Figure 1 shows the study design and flowchart for patient selection. We identified 243,156 newly diagnosed patients with newly diagnosed ESRD (ICD-9 CM: 586; ICD-10: N18.6; and were registered in the RICD database) and the patients all underwent chronic hemodialysis or peritoneal dialysis for more than 3 months.”.
For the description of “age” on figure 3, we added “Subgroup analyses (Figure 3) revealed that the ICM-CT cohort, compared with the non-ICM cohort, had a significantly higher risk of mortality regardless of range of ages on initiation of dialysis, gender, dialysis vintage (>1 year), and diabetic status.” In the paragraph of sensitivity and subgroup analyses in the result section.
- For age as an important CVD risk factor, we have added these descriptions in the result paragraph “Table 1 displays the distribution of age, sex, and comorbidities of the ICM-CT and non-ICM CT cohorts. The mean age of dialysis in the ICM-CT group was 57.2(±15.7) years and the non ICM CT group was 7(±12.8) years before propensity score matching (PSM). After PSM, the mean age of dialysis in the ICM-CT group was 63.1(±16.7) years and the non ICM CT group was 55.0(±15.6) years”. In the first non-matching model, the age in the ICM-CT exposure group was younger; then to balance the baseline risks, we extended the second propensity score matched (PSM) model. In the PSM mode, the mean age in the ICM-CT group became older than the non ICM-CT cohort. In table 2, either the crude and the adjusted model (PSM) revealed an increased risk of mortality and MACEs in the ICM-CT cohort. To further compare the contributing effect of different CVD risk factors, we analyzed the risk stratified by these risk factors (in figure 3) including range of ages on initiation of dialysis. In each age range, the outcome risk in the ICM-CT cohort in each age range was higher than that in the non OCM-CT cohort.
The results are set out and described in a great way. And it is in this part that I have a couple of comments.
Great discussion, and a notable research limitations section.
